# Nursing Ethical Decision Making on Adult Physical Restraint: A Scoping Review

**DOI:** 10.3390/ijerph21010075

**Published:** 2024-01-09

**Authors:** Vanessa Sofia Jorge Cortinhal, Ana Sofia Castro Correia, Sérgio Joaquim Deodato Fernandes

**Affiliations:** 1Centro Hospitalar Barreiro Montijo, EPE, Institute of Health Sciences, Universidade Católica Portuguesa, 1649-023 Lisboa, Portugal; 2Centro Hospitalar de Lisboa Ocidental, EPE, Institute of Health Sciences, Universidade Católica Portuguesa, 1649-023 Lisboa, Portugal; 3School of Nursing, Institute of Health Sciences, Universidade Católica Portuguesa,1649-023 Lisbon, Portugal; 4Center for Interdisciplinary Research in Health (CIIS), Institute of Health Sciences, Universidade Católica Portuguesa, 1649-023 Lisboa, Portugal

**Keywords:** physical restraint, nursing, hospitalisation, ethical decision making

## Abstract

Objective: to map the existing knowledge on nursing ethical decision making in the physical restraint of hospitalised adults. (1) Background: physical restraint is a technique that conditions the free movement of the body, with risks and benefits. The prevalence of physical restraint in healthcare suffers a wide variation, considering the environment or pathology, and it raises ethical issues that hinders decision making. This article intends to analyse and discuss this problem, starting from a literature review that will provoke a grounded discussion on the ethical and legal aspects. Inclusion criteria are: studies on physical restraint (C) and ethical nursing decision making (C) in hospitalized adults (P); (2) methods: a three-step search strategy was used according to the JBI. The databases consulted were CINAHL Plus with Full Text (EBSCOhost), MEDLINE Full Text (EBSCOhost), Nursing and Allied Health Collection: Comprehensive and Cochrane Database of Systematic Reviews (by Cochrane Library, RCAAP and Google Scholar. All articles were analysed by two independent reviewers; (3) results: according to the inclusion criteria, 18 articles were included. The categories that influence ethical decision in nursing are: consequence of the decision, the context, the nature of the decision in terms of its complexity, the principles of the ethical decision in nursing, ethical issues and universal values; (4) conclusions: the findings of this review provide evidence that there is extensive knowledge regarding nursing ethical decision making in adult physical restriction, also, it is considered an ethical issue with many associated assumptions. In this article we aim to confront all these issues from a legal perspective.

## 1. Introduction

According to Bleijlevens and colleagues (2016, p. 2309) [1] physical restraint constitutes “actions or procedures that prevent a person’s free body movement to a position of choice and/or normal access to his/her body by any manual method, physical or mechanical device, material, or equipment attached or adjacent to a person’s body that a person cannot control or remove easily” [1]. Such materials or equipment may include sheet immobilisers, gloves, splints, waistcoats, belts, wheelchair brakes, wrist bands, or immobilisers [2].

In Europe it is estimated that 6% to 85% of patients are physically restrained, and 9% to 64% in the United States [3]. The prevalence of physical restraint varies greatly depending on the environment or pathology. In nursing homes, a study found a 52% incidence of physically restrained patients [3], another study on intensive care units concluded that 0% to 100% of patients are physically restrained [4]. In a study by De Bellis and colleagues (2013) 10 articles were analysed where the population had dementia, concluding that 12% to 56% of people were physically restricted [5].

According to Chien and colleagues 2007, nurses are the health professionals most involved in making the decision to physically restrain a person as well as in the use of physical restraint in clinical practice [6,7]. Decision making according to Deodato, (2008, p. 30) is “*a process that precedes the act to respond to problems or ethical dilemmas that arise in the course of the nurse’s professional practice*” [8]. This set of phases is described as complex [9,10,11], this issue arises because physical restriction has a potential associated problem [5] and because of the available evidence not guaranteeing the prevention of accidents or accidental exteriorisation of devices. On the other hand, it is also questioned whether the practice is ethically acceptable [12].

The potential problems in people undergoing physical restraint may be physical or psychological. Studies show anxiety, fear, haematoma, oedema, change in pulse and temperature, increased capillary refill time, movement and colour, and limb ischaemia [13,14,15,16,17,18]. Some studies also delve into the implications that immobilising people has on nurses, such as psychological and moral issues [19,20]. In view of the complications presented and as the practice is considered ethically questionable, it is important to contextualise some of the reasons that studies note for the physical restraint of adults. Several authors state that the technique is used to ensure the safety of the patient or third parties, particularly in situations of agitation or aggressiveness [19,20,21]. Nurses use physical restraint frequently to prevent the risk of falling [22,23,24,25] or prevent the exteriorisation of devices such as nasogastric tubes, orotracheal tubes, or catheters [6,23,24,25]. Wang et al. (2020) in their study further explore some intrinsic factors such as age or pathologies such as dementia, which increase the likelihood of the person being physically restrained [5,6,26].

After framing the reasons that make nurses decide to physically restrain an adult patient and the main complications of this procedure, it is relevant to contextualise the ethical dimension. In this way, we can clarify all dimensions involved in decision making and justify its complexity [9,10,11]. 

One of the ethical principles that is questionable is the respect for autonomy, since the freedom of the person is always limited in the physical restriction. On the other hand, many times the person cannot consent to the procedure and a legal guardian is required [12,26]. As with all health procedures, informed consent must always be obtained, and physical restraint is no exception. It can only be waived in emergency situations. This consent must be given by the person themselves or by the legal guardian in case of incapacity of the person. If this is not the case, the decision must be made based on the greatest benefit to the person. Therefore, ethically, even if the person is externalising devices, one should not immobilise without their consent because the procedure in itself is already a restriction on their freedom [7,12,25,27]. Physical restraint should be discussed in the multidisciplinary team to reach the best possible decision [12].

In 2022, a retrospective cohort study was carried out to understand physical restriction, pre and post the COVID-19 pandemic, and it seems that there is no clarity in the results, with the prevalence increasing in some and remaining similar in others. According to this study, it seems that the phenomenon was not that influenced by the COVID-19 pandemic [28].

Nurses also feel that given the risks and associated complications, they may not be thinking of the greatest benefit to the person [7,29], thus calling into question the ethical principle of beneficence and non-maleficence. Faced with this knowledge, decision making leads us to an ethical dilemma [9,27,30]. An ethical dilemma is defined by a situation that presents two contradictory solutions [31]. Deodato has replaced the concept of ethical dilemma by ethical care, which is conceptualised by “a situation that is difficult to approach, possibly new to those facing it, but whose solution is found within ethical principles and professional values, through ethical reflection” [8].

The objective of this study is to map the existing knowledge on nursing ethical decision making in physical restraint of hospitalised adults. A search was initially conducted in JBI Evidence Synthesis, MEDLINE, and CINAHL in December 2021 in order to confirm the non-existence and relevance of this review. The studies reveal that nurses have a good level of knowledge about physical restraint, but negative practices and difficulty in attitude [32,33]. In order to minimise or eliminate physical constraint, several countries are adapting training programmes [5,34,35].

Review Question

What is the available knowledge in the scientific nursing literature on ethical decision making by nurses in the physical restraint of hospitalised adults?

Inclusion Criteria

Participants

This review included all studies that involved adults aged 18 years or older who were physically restrained. This study excludes adult pregnant women and studies with a population under the age of 18 years.

Concept

Studies addressing the topic of physical restraint, namely, ethical decision making in nursing, were considered.

Context

All studies available in the literature that include hospitalised adult who were physically restrained, excluding short- or long-term hospitalizations in nursing homes.

Types of Studies

All methods were included in this study, studies of a qualitative, quantitative, or mixed nature were accepted, including opinion articles. Unpublished studies from the “grey” literature were also included if they met the previously stated inclusion criteria.

## 2. Materials and Methods

This scoping review was conducted according to the Joanna Briggs Institute 2020 (JBI) method [36].

Search Strategy

The search strategy allowed to find published or unpublished primary studies such as literature reviews and opinion articles. The search strategy used was ordered in three steps as suggested in the JBI [36]. A limited search was initiated in the PubMed and CINAHL databases to identify relevant studies for this review and the keywords present in the title and abstract. The following research equation was obtained: [3]. The following databases were consulted: CINAHL Plus with Full Text (EBSCOhost), MEDLINE Full Text (EBSCOhost), *Nursing and Allied Health Collection: Comprehensive*, and *Cochrane Database of Systematic Reviews* (by the Cochrane Library). The grey literature was accessed through RCAAP and Google Scholar, with the keywords physical restriction, immobilisation, and nursing. Finally, in the third step, all the bibliographic references of the selected articles were analysed. No time window was applied, and articles published in Portuguese, English, and/or Spanish were included. Appendix A presents a table describing the search strategy applied in PubMed.

Study Selection

Studies were selected through analysis of the titles and abstracts by two independent reviewers, in order to ensure compliance with the inclusion criteria and to select the texts for full analysis. In the case of disagreement regarding the inclusion of studies, the reviewers discussed the case with a third independent reviewer. This selection was performed using the Rayyan Intelligent Systematic Review software 5 (Rayyan Systems Inc., Cambridge, MA, USA). Subsequently, the selected full studies were collected in Mendeley, software v1.63.0 (Mendeley Ltd., Elsevier, London, UK).

Data Extraction

Data collected from the articles answering the research question and providing information on the studies were included in a data extraction tool built by the authors. The data extraction was performed by two independent reviewers. The extracted data contains information on the author, title, objectives, year of publication, review question, country, concept, context and population, methodology, and main outcome. The data extraction tool was based on the methodological manual for scoping reviews by the Joanna Briggs Institute 2020 [36].

Data Analysis and Presentation

The studies were analysed to answer the research question and meet the objectives outlined. To clarify the extracted information, we performed a categorization through the analysis technique “content analysis” [37] using the Nvivo 14 Software (for Australia). The results will be presented in tables, based on the data extraction tool. This instrument will include the study title, objectives, concept, and main results. To allow for the interpretation of results, a relationship between the review question, objective, and the results is created in a descriptive summary.

## 3. Results

Search Results

In the present review, a total of 191 articles were initially identified through the search equation described above. After analysing the inclusion criteria, only 17 articles met the protocol. Through the bibliographic references of the included studies, 3 studies were analysed and only 1 study met the inclusion criteria, thus leaving a total of 18 studies included. The search results and the selection process are presented in the Prisma flow diagram (PRISMA-ScR) in Figure 1.

The selected articles were organised in a table by author, year, country, and method (Appendix B). The analysis of this table allows us to understand that there has been a concern with this topic since 2004. The most recent included studies were issued in 2021, which proves the thematic contemporaneity. As for the geographic distribution, the studies are located in Asia, Europe, the Americas, and Australia, which proves the thematic universality. Belgium is the country which stands out the most, with three articles meeting the inclusion criteria, followed by China, Japan, and Iran with two articles.

Study Inclusion

As for the method used and in order to assess the methodological quality of the articles, we used Melnyk’s levels of evidence, as advised by the JBI. The articles have methodological quality, as there are three non-randomised experimental articles—Level 3 of evidence; three case studies, two grounded theory, five qualitative studies that do not specify the method, one systematic literature review, one phenomenological study—Level 5 of evidence; finally, we also included one mixed study. The sample of the studies also varies, i.e., some studies allow us to check their validity, while others do not have a clear or significant sample in the article. This instrument is presented in Appendix B.

Review Findings

In order to answer the review question and its objective, the results were grouped so as to simplify their analysis. In this way, the results are categorized according to Bardin’s content analysis [38].

This analysis was performed with four previously chosen categories, the remaining categories were selected after the first text analysis. The categories that influence the ethical decision in nursing are the consequences of the decision; context; nature of the decision as to complexity; principles and universal values of ethical decision making in nursing; ethical issue; and the universal values. This analysis as well as the corresponding subcategories can be seen in Table 1.

Category:

Consequences of the decision

In the studies, it became clear that nurses’ major concern is to maintain patient safety [7,33,38,39,40,41,42,43,44,45,46,48,49,50,51,52,53,54,55,56]. This concern leads to physical restriction, but there are associated consequences for the person [7,33,38,39,41,44,46,48,49,50,51,52,57], the family [11,38]; and the nurse him/herself [7,11,33,38,41,43,44,46,48,50,55,56].

The consequences identified in the patient are physical, psychological, and social. The first may be: increased blood pressure, heart rate, and temperature; changes in the skin (haematomas, oedema) and circulation, which may lead to limb ischaemia; pressure ulcers; aspiration; pain; fractures; bladder and faecal incontinence; dehydration; urinary tract infections and respiratory infections; and death [7,33,38,39,44,46,47,48,51,52,55,57,58]. The psychological ones are depression, anger, loss of autonomy, loss of dignity, decrease in self-confidence, change in body image, fear, anxiety, aggressiveness, delirium, agitation, risk of post-traumatic disorder, confusion, and distress [7,38,39,46,48,51,55]. At last, social consequences are described as social isolation [48,55,59] and sense of abandonment [44]. In opposition there are studies that speak of the consequences of not physically restraining the patient, identifying death and increased length of stay as the main ones, associated with falls, trauma, pressure ulcers, removal of wires or tubes and, therefore, increased hospitalization costs per patient [39,42,48]. Cheug and colleagues present a study by Robbins et al. in which mortality and morbidity are eight times higher in patients immobilized in bed [44]. Considering the data presented, there are documented consequences in physically restricted people as well as in patients without any physical restraint. As regards the consequences of the decision on the family, only two articles on this topic were included. However, there is clear family suffering and the memories of this event seem to be greater in this population [11,38].

The identified articles also report harm in the population of nurses who practice this intervention reporting feelings of frustration, ambivalence, guilt, anxiety, physical problems (headache, fatigue, and gastrointestinal changes), insomnia, sadness, emotional instability, fear, anger, pity, absenteeism at work, distress, compassion, burnout, emotional, and moral distress [7,11,38,41,43,46,48,50,55,56,60]. These negative feelings experienced by nurses may contribute to errors in clinical judgment [41], affect professional practice [46,48], and even lead to ethical suffering caused by the ethical care experienced [48]. Some studies did not mention negative feelings, such as guilt or other emotional changes, and these feelings are closely related to the knowledge about the procedure. These nurses believe that their interventions are beneficial for the patient, but, on the other hand, they also do not reflect much on the action, seeing the practice as a routine [7,43,50].

Context

The context may play a decisive and even constructive role in ethical decision making in nursing [7,11,33,38,43,46,48,54,55]. Factors include the location of the patient, which affects their safety, working hours (night shift and weekends, less capacity for supervision due to a smaller number of nurses), visiting hours, or other complementary means of diagnosis because the presence of other people allows for divided supervision, shortage of time for ethical decision making in nursing and for reflection, capacity of the ward, number of hours of care, lack of health professionals, emergencies, and a lack of alternative equipment for physical restriction weight in the decision making process [7,11,33,38,41,46,48,54,55]. Another issue that arose from the results related to context is insufficient knowledge about ethics or legislation [7,11]. Regulation is also a prominent topic in this category, with studies reporting a lack of institutional guidelines on policies and ethics leaving professionals fearful of reprisals and facing ethical issues with few resources [7,41,43,46,48,54].

Studies show that by sharing the decision with other healthcare professionals or family, the consequences, previously presented in the nurse, are reduced [11,39,43,48,49,50,54] and there are cases in which other alternatives are chosen [51]. In the study by Casterlé and colleagues. this influence on ethical decision making in nursing, could m be the nurse postpones and adapts to another’s decision, discusses with another colleague, or a consensus is found involving the person’s caregivers [54]. If some of the factors indicated in the contexts were to improve, it could facilitate patient supervision, thereby decreasing physical restriction and allowing time for better ethical reflection [11]. These assumptions make nurses ignore the principles of ethical decision making [46,48].

Nature of the Decision as to Complexity

All articles included describe the nature of the decision regarding physical restraint a complex decision [11,38,48,52,54].

Principles and Universal Values of Ethical Decision in Nursing

The ethical decision in nursing regarding physical restraint is guided by the principles of ethical decision making which, as we can see in Table 1, is the theme that authors most address [7,11,33,43,44,46,48,49,50,51,52,54,55]. The nurse when deciding should reflect on the advantages and disadvantages of this procedure to be able to make an ethical decision in nursing [48]. But, in fact, during the physical restraint of the patient, the principle of autonomy is limited (acquiring informed consent by the person is sometimes impossible) [49] by the principle of beneficence and non-maleficence, because nurses prefer the patient’s safety to their freedom, feelings, or comfort [7,33,38,43,46,48,49,52,55,61]. However, in the qualitative study by Goethals et al., the nurses interviewed recognised the importance of freedom of movement [52]. Another ethical principle involved in decision making in nursing is justice, which is also broken by the lack of knowledge, with patients suffering prejudice and injustice [55].

The results also show in addition to ethical principles, there is an inherent respect for universal values, such as human rights, referring to the respect for human dignity (International Council of Nurses 2021). In the study by Salehi and colleagues 2020, the nurses interviewed reported that they felt emotional distress as they violated the patient’s rights, namely their human dignity [48]. In line with this idea, there are seven more studies included that refer to this violation of human rights when they physically immobilise the patient [33,38,43,48,49,55], also referring to the violation of human dignity [7,38,43,49], and the right to equality [55].

Ethical Issue

The results included in this category rise from the absent application of the principles and universal values when deciding to restrain a patient [11,33,38,41,43,44,48,50,52,54]. According to Goethals and colleagues, the decision whether to physically restrain the patient or not is always an ethical issue for nurses [11]. As a way to solve the problem, in one study nurses ignored ethical principles or other reflections [48], which is in agreement with another study, in which half of the nurses also report that there is no ethical issue because they perform the practice for the safety of patients [7]. Yamamoto et al. present two studies by Crisham, which states that this decision making can be influenced by education, life, and professional experience [41], and moral values [11,41].

## 4. Discussion

This review aimed to map the existing knowledge on nursing ethical decision making in physical restraint of hospitalised adults. The results were grouped in five categories, namely consequences of the decision; context; nature of the decision as to complexity; principles and universal values of ethical decision making in nursing; and ethical issues. As we stated in Section 3 the category “consequences of the decision” was subcategorized in “consequences to the person”, “consequences to the nurse”, and “consequences to the family”.

Ethical considerations on physical restraint may be limited to reflection of harms and benefits, respect for autonomy, and universal values such as human dignity [11,38,43,44,46,48,52,55]. In practice, the main conflict is between safety versus freedom of movement. After our analysis, it is clear that one of the nurses’ concerns is related to the patient’s right to safety and its limitations. The Portuguese Basic Health Law states that all individuals have the right to access healthcare appropriate to their situation, promptly and within clinically acceptable timeframes, in a dignified manner, in accordance with the best available scientific evidence and following good practices in health quality and safety. It also avowed that all individuals have the right to decide, freely and informedly, at any time, about the care that is proposed to them, except in exceptional cases provided for by law, to issue advance directives of will, and to appoint a healthcare proxy. This foundational law also states that the individual should be a part of health decision-making processes [62]. This law refers to the patient’s right to quality of care and safety practices, and we note that it does not only refer to keeping the patient safe, but also to a humanized way of maintaining their safety. This allows for respecting patients’ human dignity and considering the patient as the person who decides and intervenes in their health. The issue that prevails is if physical restraint allows for the respect of human dignity. In the physical restraint of a hospitalized person, respect for the person’s dignity must be considered, and there are ethical limits to the actions of health professionals to maintain people’s safety. As stated in the results, we found that the nurse struggles with the identification of those limits. Respect for the dignity of the human person constitutes an essential limit for all human action. Article 1 of the Charter of Fundamental Rights of the European Union [63] refers to this principle as inviolable, therefore, it must be respected and protected in all circumstances, including in healthcare.

With this review we categorized physical restriction consequences for the patient, at a physical, psychological, or social level, and also for the nurse, as a professional, and the family members. This fact contrasts with Article 3 of the Charter of Fundamental Rights of the European Union, which states that “*Everyone has the right to respect for his or her physical and mental integrity (…).*” It also consecrates that “*In the fields of medicine and biology, the following must be respected in particular: the free and informed consent of the person concerned, according to the procedures laid down by law (…)*” [64]. So, if there is clear damage to the person on a physical and psychological level, we conclude that there is a grounded disrespect for the physical and psychological integrity of the person, incurring in a violation of fundamental rights. Article 3 of the Charter of Fundamental Rights of the European Union [63] also emphasizes the importance of the person’s free and informed consent, in which the results demonstrate a contradiction with what is indicated in the legislation. According to the authors, there is a clear non-compliance on the part of nurses regarding the patient’s fundamental rights, which are regulated at various levels: institutional, national, European, and International.

As we have already presented the patient’s autonomy is not respected, namely by the non-consent of the patient for the decision to physically restrain, and the practice may then become illegal from an ethical and legal point of view [46,48,51,55]. Based on the literature that we used to sustain this study (Section 1), we consider physical restraint as a deprivation of liberty, an undignified practice, and inhumane treatment. This conclusion is based on a conceptual analysis of the very definition of physical restriction, which is largely accepted by the scientific community. Clearly, we perceive that there is a clear deprivation of freedom, when the articles mention actions or procedures that prevent “free movement” or “normal access to the body”. Therefore, the person deprived of such liberty must be informed, and allowed to consent, freely and knowledgeably, at the moment that precedes any healthcare intervention related to physical restraint. Regarding the content of such information preceding the act, the person must be correctly informed about the objective, risks, and consequences of the intervention and, in this way, have freedom of choice.

There is a limit very well defined by Article 4 of the Charter of Fundamental Rights of the European Union that prohibits inhuman or degrading treatment and acts of torture, here we realize that, once again, there is a risk for incurring in a violation of the law [63]. The European Union requires a high level of protection of human health and enshrines this right in Article 35 of the Charter of Fundamental Rights of the European Union [64]. Under this legislation, any citizen eligible for this protection is safeguarded in this context at national and international level. We can consider two ways of looking at this correctly. On the one hand, preventing any harm to the patient through physical restraint may protect the person’s health, but on the other hand, it may also be considered an inhumane and cruel treatment. Patients have rights and duties, but health professionals also have obligations and well-defined rules of conduct from an ethical and legal point of view, and for which they are responsible for; this affirmation is sustained by Article 4 of the “Convention for the Protection of Human Rights and Human Dignity in Regard to Applications in Biology and Medicine: Convention on Human Rights and Biomedicine” (Resolution of the Assembly of the Republic No 1 of 3 January. Convention for the Protection of Human Rights and Human Dignity in Regard to Applications in Biology and Medicine: Convention on Human Rights and Biomedicine 2001).

Our major limitation is the evidence level of our included studies. There were only studies with level of evidence from tier three down, according to Melnyk, i.e., the samples are always intentionally chosen. This can report the lower level of evidence that sustains our interpretations but also shows the lack of investment in studying this common practice that by nature constitutes a complex decision to make as an individual professional or as a professional in a healthcare team environment, and very frequently is presented as an ethical issue. One of the topics on which there is no conflict between the authors is the nature of the decision as to its complexity, who clearly define it as complex. Dynamic decision making is very context-dependent, but it deserves to be highlighted due to the relevance assigned to it by the studies.

Another limitation of our study is that most included studies were conducted in intensive care units (ICUs). All nurses are part of a context, but only studies in which the person is hospitalised were included in this study. Although we choose to not include psychiatric contexts or nursing home institutions in the sample for purposes of studying very clearly what we considered to be our focus: hospitalized adult (considering the adult without diagnosed psychiatric illnesses), this fact shows the absence of studies focused in populations that are admitted in emergency rooms or admitted for treatment in other settings of adult hospitalized care, like surgical post-operative recovery floors. This indicates that the concerns, strategies and knowledge we collected is conditioned by a specific nurse-patient ratio and a certain setting of care, this limits the applicability of our study to the various contexts of nursing practice.

Further studies should aim for higher levels of evidence and search for implications of physical restraint in hospitalized adults in settings other than ICU’s. Future researches should aim to deepen the knowledge into dynamic decision making regarding physical restraint in a multidisciplinary healthcare team.

## 5. Conclusions

The results of this review clearly show that there is reported knowledge about ethical decision making in nursing in the adult physical restriction, and also it is clearly identified as an ethical issue with many associated assumptions. These data contribute to improved knowledge in this area for nurses, but there are several described consequences for the person, family, and nurse. By analysing the studies, ethical decision making in this case is a balance between ethical values, universal values, and the value of safety [7,11,46,48,49,50,51,55]. What is still open is how to achieve this balance and whether the greatest benefit of physical restriction will effectively be for the patient or for the nurse [7]. 

We believe that randomised experimental studies should be conducted in the future so as to increase knowledge and validate nursing ethical decision making in this area. Studies are also needed to characterise the phenomenon from the perspective of the patient and family, as well as to extend the study to different contexts.

## Figures and Tables

**Figure 1 ijerph-21-00075-f001:**
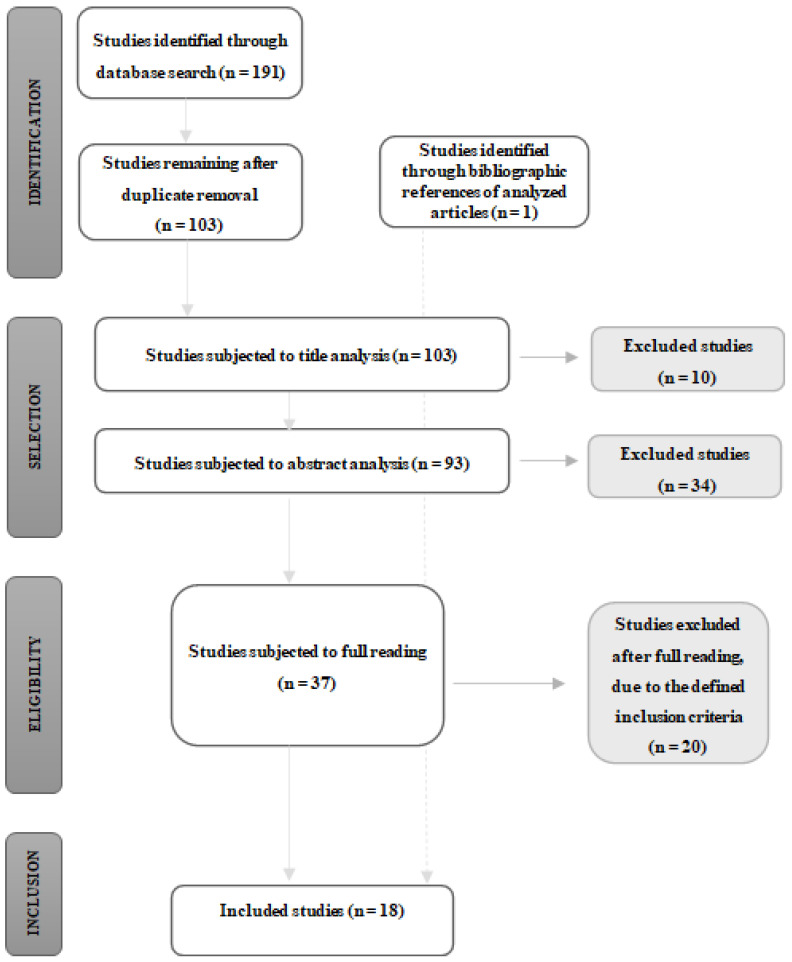
Results flowchart adapted from that proposed in the methodological manual for scoping reviews of the Joanna Briggs Institute.

**Table 1 ijerph-21-00075-t001:** Nursing ethical decision making on physical restraint in the adult using Bardin’s content analysis, 2011 [39].

Category	Subcategory	Study Count	Reference Count
Consequences of the decision	**Consequences of the decision on the person** Physical: Increased blood pressure, heart rate, and temperature; changes in skin (bruising, oedema) and circulation can lead to limb ischemia; pressure ulcers; aspiration; pain; fractures; bladder, and faecal incontinence; death; dehydration; urinary tract, and respiratory infections [33,38,39]; Psychological: Depression, anger, loss of autonomy, dignity violated, decreased self-confidence, altered body image, fear, anxiety, aggression, delirium, agitation, risk of post-traumatic disorder, confusion, and distress [7,39,40,41,42]; Social: Social isolation [39,40,43]; and sense of abandonment [44].	13	42
	**Consequences of the decision on the nurse** Frustration, ambivalence, guilt, anxiety, physical problems (headache, fatigue, and gastrointestinal changes), insomnia, sadness, emotional instability, fear, anger, pity, absenteeism from work, anguish, compassion, burnout, emotional, and moral distress [7,39,40,45,46,47].	12	40
	**Consequences of the decision on the family** Family suffering [11,48].	2	3
		10	31
Context	Inadequate knowledge	2	3
	Regulations	6	12
	Dynamic decision making	7	14
Nature of the decision as to complexity	Complex decision	5	10
Principles and universal values of ethical decision making in nursing	Autonomy; justice; beneficence; non-maleficence. Human dignity [7,38,39,47,48,49]; right to equality [40].	15	65
Ethical issue	Ethical issue	13	30

## Data Availability

Supporting data for the findings of this research are available on request.

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
