# Peer review of "Nursing Ethical Decision Making on Adult Physical Restraint: A Scoping Review"

_ijerph, 2024, doi:10.3390/ijerph21010075_

Round 1

Reviewer 1 Report

Comments and Suggestions for Authors

Congratulations to the authors on the excellent scientific work produced. However, there are some points that may need further attention to make the work more consistent:

  1. Formatting errors in the text, specifically at line 156.
  2. Spelling mistakes, for example, it should be "Prism" not "Prisma" on line 166.
  3. The formatting diverges from what is outlined in the journal's guidelines: "References should be numbered in order of appearance and indicated by a numeral or numerals in square brackets—e.g., [1] or [2,3], or [4–6]."
  4. The DOI numbers are completely missing.
  5. The PRISMA guidelines require additional points for scoping reviews, such as limitations. For more, see: https://knowledgetranslation.net/wp-content/uploads/2019/05/PRISMA-ScR_TipSheet_Item20.pdf

I look forward to your evaluation. Best wishes and good work.

Author Response

Good afternoon, Reviewer 1,

Firstly, I appreciate all your suggestions, as they have undeniably played a crucial role in enhancing the quality of this article. I will now address each point:

1 and 2. These have been corrected in the text;

3 and 4. I have revisited all the references, and I believe they now align with the journal's guidelines, including the inclusion of DOIs;

5. This is addressed starting from line 442 and has been inserted into the discussion, as exemplified.

I am available for any further questions.

King regards

Vanessa Cortinhal

Reviewer 2 Report

Comments and Suggestions for Authors

Congratulations for your paper. Thank you for the opportunity to review your work. This paper examines 18 articles in the field of ethical decision making on adult physical restraint.

This paper is focused on an important and timely topic that is relevant to an international academic audience, and has significant relevance for a wider public. Some of the strengths of this work include clear writing and construction as well as appropriate use of and discussion of methods. However, there are also some changes and additions that would increase the impact and significance of the work.  Address the suitability of the article for the journal's scope and audience, also keeping in mind that the journal’s readership is international, European Union laws in more suitable them Portuguese law. Adding more substantive discussion around future directions would be a substantive and valuable addition.

I really like this article, but there is a polishing that needs to be done and questions that need to be answered:

Line 33 - and colleagues not et al.

Line 43/44 - and colleagues not et al.

Line 46 - and colleagues not et al.

Line 110 – 112 Context – hospitalized adults it is a range of settings, can you explain? In background was present evidence from nursing homes, when do you refer hospitalized, a patient in nursing home are hospitalized too? Ethical issues in decision making maybe will be very different in a hospital setting in acute care or in long term care.

Line 125 -The research equation just search in Title why you restrict to TI? This is a limitation?

Line 131 – Do you consider to search in open grey?

Line 155 - 159 format text

Line 170 -Why do you consider that are still issues to be studied in this field (ethical decision making)? If you consider this maybe you need to explain this sentence.

In methods it is not clear the protocol. It will be great if you indicate whether a review protocol exists; state if and where it can be accessed (e.g., a Web address); and if available, provide registration information, including the registration number.

Eligibility criteria is other item need to be detailed, e.g. specify characteristics of the sources of evidence used as eligibility criteria (e.g., years considered, language, and publication status), and provide a rationale.

In general, the discussion section needs to improve to fix the goals proposed for JBI to ScR, namely you need to summarize the main results (including an overview of concepts, themes, and types of evidence available), link to the review questions and objectives, and consider the relevance to key groups. Discuss with more accuracy the limitations of the scoping review process. And need to improve the general interpretation of the results with respect to the review questions and objectives, as well as potential implications and/or next steps.

References – without the 18 references included in the results of the ScR, the majority are with more than 5 years. You need to improve the overall references.

Author Response

Good afternoon, Reviewer 2,

First and foremost, I want to express my gratitude for your invaluable suggestions, which undeniably played a crucial role in enhancing the quality of this article. I will now address each point:

  1. All issues regarding "and colleagues, not et al." have been resolved in the text;

  2. Regarding the context, the decision to focus on the hospital as the area of interest was made by the researchers. While there are some references to homes in the introduction for contextualization purposes, it's important to note that this would be a subject for a separate study;

  3. Regarding the research equation, it was an investigator's choice that we tested, and it does not appear to impose limitations on the study;

  4. The research included a thorough examination of gray literature, and no relevant results were found;

  5. In response to the concern raised in Line 170, I have removed the statement;

  6. There is no published review protocol;

  7. Regarding the eligibility criteria, all relevant information is available in Appendix 2, easily accessible for consultation. Would it be clearer to incorporate this information into the main text?

  8. The entire discussion has been thoroughly reviewed, and all changes are highlighted. I trust that these revisions address the suggestions made;

  9. To enhance the references, we reran the search using the same equation up to the date of completion. Unfortunately, no new relevant results were found. The initial search yielded 11 articles, of which 4 were included. One article was inaccessible, and after analysis, it did not contribute new knowledge. Additionally, we took the opportunity to update information in the introduction to include developments up to 2022.

I am available for any further clarifications or questions.

Best regards 

Vanessa Cortinhal

Reviewer 3 Report

Comments and Suggestions for Authors

As the authors express clearly, the study aims to map the knowledge on nursing ethical decision-making in physical restraint of hospitalised adults.

The study seems generally well-written, and the content is interesting to read.

I do not find essential errors, but all non-English text, incl. the references, should also be translated into English.

There are methodological and content issues to discuss, which may be considered included.  

Regarding the method scoping review:

This method should be better discussed, including what scoping review is not, compared to other reviews, e.g. meta-analysis. It's important to mention the limitations of the method. Examples of limitations are the lack of a detailed assessment of the quality of the cited studies, - and what about this method and the risk of bias? It is also important to note that this method is typically broad at the expense of depth. An instruction guide for this method mentions that this method should include hand-searching the literature. Is this issue of relevance here – or why not?

This paper is about NURSING ethics and decisions: 

The mentioned ethics discussed is very much (too much?) about doctor/medical ethics, although the concepts of care and phenomenology are fortunately included. But these concepts are rather narrowly discussed. There is little about virtue ethics,  e.g. not much about the traditional nurse ethics of compassion, inspired by the famous and traditionally much-used parable about The Good Samaritan (which is discussed, e.g. in the Scandinavian phenomenological nurse ethics tradition, as K.E. Løgstrup, Kari Martinsen).  

The title stresses that the paper is about decisions, but the discussion of this concept seems rather narrow, without mentioning the massive field of decision theory. The paper cites several times the word consequences, but in ethics, consequences should be discussed from various perspectives, e.g.  as in utilitarianism, and ethical decisions should, according to many ethicists, also be judged according to other dimensions than consequences, as in intention ethics (Kantian ethics, etc.).

Regarding physical restraint:

This concept of restraint is so much used in the psychiatric field of medicine that it would be useful to include the knowledge and experiences from this field, i.e., more than just mentioning this area passively in tables and references. 

Comments on the Quality of English Language

Translate non-English text.

Author Response

Good afternoon, Reviewer 3,

First and foremost, I want to express my gratitude for your invaluable suggestions, which undeniably played a crucial role in enhancing the quality of this article. I will now address each point:

  1. Regarding the methodology, it's important to note that one of the advantages of a scoping review is the inclusion of all relevant literature, as quality assessment is not a prerequisite according to JBI 2022. The manual search involved consulting the bibliographic references of each article.

  2. Concerning nursing ethics, we acknowledge the importance of the issue. However, we believe that ethics is not exclusive to nursing. The authors grounded their work in bioethical principles and current legislation. Given that the decision-making process for physical restraint involves nurses, the insights can be applied broadly to improve nursing care through enhanced decision-making.

  3. Continuing on the topic of ethics, the authors made a deliberate methodological decision due to the extensive existing knowledge on decision-making. This decision aimed at streamlining the focus and avoiding redundant information that is already widely discussed.

  4. We appreciate your feedback once again regarding the criticism related to the psychiatric population. While knowledge in psychiatry is undoubtedly valuable, the study's objective is to shed light on an equally vulnerable but non-psychiatric population.

I am available for any further clarifications or questions.

Best regards 

Vanessa Cortinhal